# The Picture Interpretation Test 360°: A Virtual Reality Screening Tool for Executive Dysfunction and Rehabilitation Stratification in Mild Cognitive Impairment

**DOI:** 10.3390/healthcare14010095

**Published:** 2025-12-31

**Authors:** Chiara Stramba-Badiale, Eleonora Noselli, Alessandra Magrelli, Silvia Serino, Chiara Pupillo, Stefano De Gaspari, Sarah Todisco, Karine Goulene, Marco Stramba-Badiale, Cosimo Tuena, Giuseppe Riva

**Affiliations:** 1Applied Technology for Neuro-Psychology Laboratory, IRCCS Istituto Auxologico Italiano, 20145 Milan, Italy; eleonora.noselli@unicatt.it (E.N.); magrelli13@gmail.com (A.M.); tod.sarah88@gmail.com (S.T.); c.tuena@auxologico.it (C.T.); giuseppe.riva@unicatt.it (G.R.); 2Humane Technology Lab, Università Cattolica del Sacro Cuore, 20123 Milan, Italy; chiara.pupillo@unicatt.it (C.P.); stefano.degaspari@phd.unipi.it (S.D.G.); 3Department of Psychology, Università degli Studi di Milano-Bicocca, 20126 Milan, Italy; silvia.serino@unimib.it; 4Bicocca Center for Applied Psychology, University of Milano-Bicocca, 20126 Milan, Italy; 5Department of Computer Science, University of Pisa, 56126 Pisa, Italy; 6Department of Medicine, Neurology and Rehabilitation, IRCCS Istituto Auxologico Italiano, 20145 Milan, Italy; goulene@auxologico.it (K.G.); stramba_badiale@auxologico.it (M.S.-B.); 7Department of Theoretical and Applied Sciences, eCampus University, 22060 Novedrate, Italy

**Keywords:** mild cognitive impairment, executive functions, virtual reality, cognitive rehabilitation, ecological assessment, personalized intervention, functional decline, Picture Interpretation Test

## Abstract

**Background/Objectives:** Mild Cognitive Impairment (MCI) represents a critical transition stage between normal aging and dementia, with executive dysfunction playing a key prognostic role. Traditional neuropsychological tests show limited ecological validity and may fail to detect early executive deficits. Virtual Reality (VR) offers an innovative alternative by reproducing everyday situations in realistic environments. This study investigated whether the Picture Interpretation Test 360° (PIT 360°), a VR-based assessment, can (1) discriminate between MCI patients and healthy controls (HCs); (2) identify executive dysfunction within the MCI group; and (3) correlate with standard neuropsychological measures. **Methods:** One hundred and one participants aged ≥65 years (53 MCI, 48 HCs) underwent a comprehensive neuropsychological assessment and PIT 360° evaluation. The PIT 360° requires interpreting a complex scene in a 360-degree virtual environment. Hierarchical linear regression, Receiver operating characteristic (ROC) curve analysis, and binary logistic regression were performed to examine group differences and diagnostic accuracy. MCI patients were stratified based on their performance on the Modified Five Point Test to identify visuospatial dysexecutive deficits. **Results:** MCI patients showed significantly longer PIT 360° completion times than HCs (92.6 vs. 65.3 s, *p* = 0.006), independent of age. MCI patients with visuospatial dysexecutive deficits exhibited the most severe deficits (median = 105 s, *p* = 0.017 vs. HCs). ROC analysis revealed adequate discriminative ability (AUC = 0.64, 95% CI [0.53, 0.75]) with a preliminary, sample-derived cut-off at ≥22 s, yielding high sensitivity (86.5%) but low specificity (42.6%). This threshold requires validation in independent samples. PIT 360° completion time correlated significantly with visuospatial executive functions, visual memory, and verbal fluency. **Conclusions:** The PIT 360° effectively screens for visuospatial executive dysfunction in MCI with high sensitivity, making it suitable for ruling out clinically significant impairment. Its ecological validity, brief administration, and correlations with traditional measures support integration into routine clinical practice for early detection and rehabilitation planning.

## 1. Introduction

Later life is often accompanied by a range of health-related challenges, and cognitive decline represents one of the most significant. In this context, Mild Cognitive Impairment (MCI) is a crucial intermediate stage within the continuum from normal aging to dementia and is recognized as a major risk factor for progression [1].

At the clinical level, although episodic memory problems are the most frequently reported by patients with MCI and the most extensively studied in the literature, attention has also been directed toward non-memory deficits, particularly those involving executive functioning [2,3,4].

Executive functions are higher-order processes that regulate goal-directed behavior and adaptation to novel situations, playing a crucial role in cognitive, emotional, and social abilities, as well as in everyday functioning [5,6]. For example, in MCI, deficits in these processes often translate into concrete difficulties with instrumental activities of daily living (IADLs), such as managing finances, preparing meals, or handling medications. Notably, impairments in complex IADLs have been identified as strong predictors of dementia onset [7]. This evidence suggests that executive decline in MCI should be regarded not only as a cognitive marker but also as a functional indicator with significant prognostic value for disease progression. Therefore, early and accurate detection of executive deficits is essential to enable timely interventions and guide targeted cognitive rehabilitation strategies.

Visuospatial executive functions, which integrate higher-order control processes such as planning, mental flexibility, and working memory with spatial processing [8], are among the executive domains most susceptible to cognitive impairment in MCI [9]. These functions refer to the ability to mentally manipulate, organize, and process spatial information in a goal-directed manner, requiring the coordinated activity of executive-control networks including the anterior insula, inferior frontal cortex, and medial prefrontal regions [10].

Recent evidence demonstrates that MCI patients show significant impairments in visuospatial executive performance compared to HCs, as measured by tasks such as design fluency, a non-verbal test assessing the ability to generate novel visual patterns under executive control [11]. Notably, deficits are observed not only in overall performance but also in strategic approach and solution generation, highlighting the early vulnerability of these complex cognitive processes in prodromal dementia stages [12].

Currently, executive functions are evaluated through paper-and-pencil neuropsychological tests, such as the Wisconsin Card Sorting Test [13], the Stroop Test [14], the Trail Making Test (TMT) [15] and the Modified Five Point Test (MFPT) [16]. In particular, the MFPT is a nonverbal design fluency task designed to identify visuospatial executive deficits. It requires individuals to initiate, generate, and monitor as many unique figural designs as possible within a specified time limit (typically 3 or 5 min).

However, these instruments present important limitations in ecological validity, as they are administered in controlled and distraction-free environments [17]. Moreover, it has been observed that they may yield apparently normal results even in patients who exhibit executive deficits in their daily lives [18], thus highlighting their limited sensitivity in the early detection of executive dysfunction. In addition, traditional neuropsychological tests are time-consuming and costly, further limiting their feasibility in routine clinical practice [17].

In this context, VR offers an innovative alternative [19]. VR encompasses a range of technologies, from non-immersive desktop systems to fully immersive Head-Mounted Displays (HMDs). For cognitive assessment purposes, immersive VR, delivered through HMDs such as the Oculus Rift or HTC Vive, provides the highest degree of ecological validity by creating a sense of presence in which users feel physically located within the virtual environment. By reproducing everyday situations within realistic three-dimensional environments, VR enhances ecological validity and generates the subjective sensation of “presence” or “being there” [20]. At the same time, this technology enables modulation of environmental distractions and detailed monitoring of patient behavior, thereby enhancing the sensitivity of assessments and providing insights that are often overlooked by conventional approaches. Importantly, these advantages can be achieved with reduced administration time and lower costs compared to traditional neuropsychological testing, further strengthening the clinical applicability of VR [20].

Among VR-based tools for executive assessment, the Picture Interpretation Test (PIT 360°) is a particularly innovative application of this technology. The PIT 360° is administered through an immersive HMD that presents a 360-degree panoramic image. Participants, seated comfortably while wearing the VR headset, use head movements to explore the entire visual field. They have to look up, down, left, and right, and to examine all elements of the scene. Unlike interactive VR applications that require hand controllers, the PIT 360° relies solely on visual exploration through head rotation, making it accessible to older adults with limited motor skills. The PIT 360° immerses patients in a realistic three-dimensional scene that they must actively explore and interpret, requiring coordination of multiple executive processes, such as visual scanning, hypothesis generation, detail integration, and interpretive selection, within 180 s. The theoretical foundation of this approach dates back to Luria’s seminal work in the 1960s with the concept of “active visual perception” [13]. According to Luria’s model, interpreting a meaningful image requires a coordinated series of actions, which include focusing on the most informative aspects of the visual array, comparing and integrating various details that are perceived, formulating theories about the scene’s overall meaning, and evaluating and selecting among conflicting interpretations [21]. Critically, this process unfolds through recursive cycles of exploration and hypothesis testing, with strategic eye movements enhancing the collection of visual information that progressively refines interpretive hypotheses. According to Luria, prefrontal dysfunctions would decrease the effectiveness of this active exploration and, as a result, reduce an individual’s ability to interpret the meaning of a complex image accurately.

To clinically assess this ability, Luria employed Ilya Repin’s picture “Unexpected Return” (1884), which shows a revolutionary returning home after incarceration.

The interpretation of this scene must be inferred by combining body language, facial expressions, and contextual information. In line with the impaired hypothesis generation and verification processes, patients with frontal lesions tended to describe specific scene aspects in fragments despite being unable to integrate these details into a coherent interpretation [21]. In fact, rather than systematically searching for critical cues and comparing them to form a coherent narrative, these patients exhibit an impulsive, reactive pattern. An isolated detail triggers an immediate interpretation, which remains disconnected from other elements. When prompted to look more carefully, they shift attention to another fragment, generating a new, equally isolated statement [21].

Rosci and colleagues [22] first modified and validated the PIT for Italian clinical populations, building on Luria’s groundbreaking theoretical and clinical work. They chose Giacomo Favretto’s 19th-century picture “Il Sorcio” (“The Mouse”), which displays a household scenario in an old country house where three terrified girls wait on chairs as a boy searches the floor, rather than Luria’s initial stimulus (Repin’s “Unexpected Return”). The mouse itself is not visible, but the girls’ looks and the boy’s exploration under an old cupboard suggest that it is there. Through their validation studies, the PIT proved to be a clinically useful screening tool for detecting visuospatial executive impairments, demonstrating particular sensitivity in patients with prefrontal lesions. Importantly, the test demonstrated sensitivity comparable to established measures while requiring minimal administration time [22].

Building on this theoretical framework and the clinical utility demonstrated by Rosci’s work, researchers later developed a VR version of the PIT to improve its ecological validity and clinical applicability further. The shift from a static 2D image to an immersive 360° VR environment offers several significant benefits.

First, the 360° format requires participants to actively explore the environment through head movements and visual scanning, rather than passively viewing a confined frame. This active exploration more closely mirrors real-world visual behavior, in which individuals must orient themselves in space and systematically search complex environments, both processes mediated by executive control.

Second, the immersive VR presentation enhances ecological validity by simulating the spatial demands of real-life scene interpretation, where relevant information is not always immediately visible within a restricted visual field. This spatial component adds an executive load that is absent in traditional 2D presentations, as participants must maintain spatial memory of previously explored areas while continuing to search for new information. Third, the VR environment enables precise behavioral monitoring that is not possible with static images, including exploration strategies and the temporal-spatial distribution of visual attention. Also, the VR adaptation maintains the essential interpretive challenge of the original task: inferring the presence of a hidden element (“the mouse”) based on contextual cues while updating the scene to a contemporary setting (a modern room rather than a 19th-century house).

The PIT 360° has been validated in two clinical populations with executive dysfunction. Realdon and colleagues [23] demonstrated its effectiveness in detecting executive deficits in Multiple Sclerosis (MS) patients that traditional neuropsychological batteries failed to identify. Serino and colleagues [24] extended validation to Parkinson’s disease (PD), showing that although interpretation success rates were similar between patients and controls, PD patients required significantly longer completion times and provided more detailed but less focused descriptions, reflecting difficulties in active visual perception. Importantly, interpretation time correlated significantly with traditional executive function tests in both studies, supporting construct validity. These studies established the PIT 360°’s clinical utility and safety profile in neurological populations, with minimal adverse events (cybersickness) and high completion rates.

Given these premises, the PIT 360° represents a promising candidate for MCI screening, as it combines ecological validity with the ability to assess complex executive functions in an immersive yet controlled environment. However, despite encouraging findings from studies demonstrating its utility as a diagnostic screening tool in other clinical populations, no research has yet examined its application specifically in individuals with MCI.

This study aims to investigate whether PIT 360° can discriminate between patients with MCI and HCs. Furthermore, by examining PIT 360° performance alongside already validated neuropsychological tests, we aim to (1) assess whether PIT 360° can discriminate between MCI patients with and without executive dysfunction, and (2) explore the relationships between PIT 360° measures and standard neuropsychological assessments of executive functions. Based on the literature, we predict: (1) MCI patients to have lower PIT 360° performance than HCs; (2) PIT 360° performance would predict MCI diagnosis; (3) PIT 360° performance is correlated with visuospatial executive tests (e.g., design fluency).

## 2. Materials and Methods

### 2.1. Participants

The sample size calculation was determined a priori using GPower (version 3.1.9.7, Heinrich-Heine-Universität Düsseldorf, Düsseldorf, Germany) for a one-way ANOVA comparing two independent groups (HCs vs. MCI). Assuming a medium effect size (f = 0.33) based on Cohen’s conventions for group differences in cognitive performance, α = 0.05, and desired power of 0.90, the analysis yielded a required total sample size of *N* = 100 participants. A total of 110 participants were initially enrolled. Nine participants were subsequently excluded after further clinical review revealed a diagnosis of PD, resulting in a final sample of 101 participants, aged 65 years and above (48 HCs, 53 MCI), recruited between 2 February 2024, and 3 March 2025, at the Outpatient Clinic of the Department of Medicine, Neurology and Rehabilitation, IRCCS Istituto Auxologico Italiano-Mosè Bianchi, Milan. Participants were recruited consecutively from the outpatient neuropsychological assessment service. MCI patients were referred by physicians (neurologists, geriatricians, or general practitioners) for cognitive evaluation. HC participants included: cognitively intact individuals accompanying patients to clinic appointments who volunteered for assessment, and community-dwelling volunteers previously screened at the clinic and recontacted for participation. All participants underwent an identical comprehensive neuropsychological evaluation. All participants provided written informed consent before enrollment. The study was approved by the Institutional Review Board (Ethics Committee approval no. 2023_01_31_11).

Participants were divided into two groups according to a priori diagnostic criteria: 53 individuals with MCI and 48 cognitively HCs. The presence of cognitive impairment was determined by an interdisciplinary team including neuropsychologists and a physician, based on patient anamnesis, medical reports, and comprehensive neuropsychological assessment.

For the HC group, participants were required to (i) be aged 65 years or above; (ii) show no evidence of cognitive impairment; (iii) maintain independence in functional abilities; and (iv) score above the Italian Mini-Mental State Examination (MMSE) cut-off for cognitive deterioration [25] adjusted for age and education: ≥22 for age 65–89 years.

For the MCI group, diagnosis followed the core clinical criteria established by Albert and colleagues [26]: (i) concern regarding cognitive changes expressed by the participant, an informant, or a clinician; (ii) objective evidence of cognitive impairment in one or more cognitive domains; (iii) preservation of independence in functional abilities; and (iv) absence of dementia. Additional requirements included (i) age 65 years or above; and (ii) MMSE score above the cut-off for cognitive deterioration (3).

To explore potential heterogeneity within the MCI group, patients were subsequently stratified based on the presence of visuospatial dysexecutive deficits, as assessed by the MFPT-UD. This stratification was performed post hoc for exploratory analyses (see Section 2.4).

Exclusion criteria applied to all participants included: (i) acute stroke or transient ischemic attack within the previous 6 months; (ii) neurological conditions (e.g., MS, PD) or psychiatric disorders (e.g., schizophrenia), as well as untreated mood or anxiety disorders; (iii) history of traumatic brain injury with loss of consciousness; (iv) physical or functional impairments (e.g., musculoskeletal disorders, limb paresis or paresthesia) that could interfere with VR use or experimental procedures; (v) uncorrected visual impairment; and (vi) self-reported recurrent vertigo.

### 2.2. Neuropsychological Assessment

All participants underwent a comprehensive neuropsychological battery designed to assess global cognition, memory, executive functions, attention, and functional status. The MMSE was administered to evaluate overall cognitive functioning [25]. The Frontal Assessment Battery (FAB) was used to examine frontal executive functions [27]. The Rey–Osterrieth Complex Figure Test (ROCF) measures visuospatial construction abilities and visual memory [28]. Participants have to copy a complex geometric figure (ROCF-C) and subsequently reproduce it from memory at two time points: immediately after a 30 s delay (ROCF-I) and after a 20 min delay (delayed recall-ROCF-D). The Prose Memory test (PM) evaluates verbal episodic memory both immediately after a 30 s delay (PM-I) and following a 20 min delay (PM-D) [28]. The TMT was administered to evaluate visual attention, processing speed, and cognitive flexibility [15]. The MFPT assesses visuospatial flexibility, with the primary score being the total number of unique designs produced (MFPT-UD) [16]. The Verbal Fluency test includes three components: Phonemic Verbal Fluency (PF), Semantic Verbal Fluency (SF), and Alternating Phonemic/Semantic Fluency (AF). This test evaluates verbal production, lexical abilities, and cognitive flexibility in switching between different subtasks [29]. The Digit Forward test (DF) assesses verbal short-term memory capacity [30]. The Corsi Forward test (CF) evaluates visuospatial short-term memory [31]. The Corsi-supra span (CSS) measures visuospatial learning capabilities [32].

While the neuropsychological battery did not include a formal assessment of pragmatic language and theory of mind, language skills were assessed informally through other verbal tasks (e.g., verbal fluency tests and prose memory). The study would not have included participants who showed significant expressive language deficits, anomia, or comprehension issues during these tasks. Furthermore, possible pragmatic communication deficits that could affect scene interpretation were identified through the neuropsychologist’s medical history interview.

To assess depressive symptoms, we used the short form of the Geriatric Depression Scale (GDS), a self-report questionnaire [33]. The Tilburg Frailty Indicator (TFI) [33] was used to evaluate the physical, psychological, and social self-reported frailty. Finally, the Activities of Daily Living (ADL) and IADL scales were used to assess functional independence [34,35].

### 2.3. PIT 360°

This is the immersive VR version of the PIT [22] environment. It is delivered through an HMD (Oculus Quest 2).

#### 2.3.1. Assessment Procedure

Participants were seated in a quiet, well-lit room and fitted with the VR headset, which was adjusted for comfort and optimal visual clarity. No hand controllers were required; participants explored the environment solely through natural head rotation, allowing them to look in any direction (up, down, left, right, behind) as they would in a real room. A research assistant remained present throughout the brief administration (maximum 180 s) to monitor for signs of discomfort or cybersickness and to record verbal responses (Figure 1).

#### 2.3.2. Virtual Environment

A single 360-degree panoramic still photograph of a contemporary domestic scene (Figure 2). The scene depicts three girls, appearing frightened, standing on chairs, while a boy searches for something on the floor near the furniture. Although the object of the search is not directly visible, contextual cues, including the characters’ fearful facial expressions, defensive body postures, and the boy’s directed searching behavior, clearly suggest the presence of a small threatening animal (a mouse) hidden from view. This digital representation is a contemporary adaptation of the 19th-century painting “Il Sorcio” (“The Mouse”) by Giacomo Favretto (1878), updated to a modern setting to enhance its relatability to current audiences.

#### 2.3.3. Task Instructions and Scoring

Participants are instructed to explore the scene and describe what is happening. The main outcome measures are (a) the time required for correct interpretation of the scene (PIT 360°_completion time, in seconds), defined as explicitly identifying or inferring the presence of the hidden mouse/small animal; and (b) the number of scene elements verbally described before reaching the correct interpretation (PIT 360°_number of elements). The maximum time allowed is 180 s, after which the task is discontinued.

### 2.4. Statistical Analysis

Of the 101 participants included in the study, one participant from the MCI group experienced cybersickness during the PIT 360° assessment and was therefore excluded from analyses involving this instrument. Consequently, PIT 360° analyses were conducted on 100 participants (48 HCs, 52 MCI), while all 101 participants completed the remaining neuropsychological assessments.

All statistical analyses were performed using R (version 3.6.3; R Core Team, 2024). Descriptive statistics, including means, standard deviations, medians, and interquartile ranges (IQR), were calculated for all variables.

Before conducting group comparisons on PIT 360° measures, data were screened for normality using Shapiro–Wilk tests and for homogeneity of variance using Levene’s tests. Due to significant violations of normality assumptions, the PIT 360° completion time was log-transformed, and the PIT 360° words produced until the correct response (n° items) was transformed with the squared root formula. This set of operations resolved issues with normality.

To examine group differences in PIT 360° measures, a hierarchical linear regression was conducted. Model 1 included Group as the sole predictor, while Model 2 added potentially significant between-groups covariate/s. The significance of adding Age was evaluated using F-tests for nested linear models and likelihood ratio tests (chi-squared) for nested logistic models. A parallel analysis was conducted for the number of naming errors.

As an exploratory analysis, to examine potential heterogeneity within the MCI group, patients were stratified based on the presence of visuospatial dysexecutive deficits, operationalized as impaired or borderline performance (≤10th percentile) on the MFPT-UD. Group comparisons across three groups (HCs, MCI without visuospatial dysexecutive deficits, MCI with visuospatial dysexecutive deficits [MCIdys]) were conducted using Kruskal–Wallis tests due to unequal sample sizes, followed by post hoc pairwise Wilcoxon rank sum tests with Bonferroni correction for multiple comparisons.

Binary logistic regression was performed to evaluate whether PIT 360° measures predicted MCI group membership (coded as 1 = MCI; 0 = HCs). A hierarchical approach was used, with Model 1 including PIT 360° measures alone and Model 2 adding potentially significant group covariate/s. Model fit was assessed using chi-square likelihood ratio tests. Odds ratios (OR) with 95% confidence intervals were calculated to quantify the strength of association.

Receiver operating characteristic (ROC) curve analysis was conducted to evaluate the discriminative ability of PIT 360° scores in distinguishing MCI from HCs. The optimal classification threshold was determined using Youden’s index, which maximizes the sum of sensitivity and specificity. Classification performance metrics included overall accuracy, sensitivity, specificity, positive predictive value (PPV), negative predictive value (NPV), balanced accuracy, and Cohen’s kappa. McNemar’s test was used to evaluate asymmetry in misclassification patterns [27]. Areas under the curve (AUC) was reported as a synthetic measure of overall discriminative performance: usually, values below 0.60 indicate a failure in diagnostic performance [36].

Pearson correlation analyses were conducted to examine relationships between PIT 360° measures and other neuropsychological measures in the full sample (*N* = 101). Correlation coefficients and associated *p*-values were reported following APA guidelines, with *p*-values less than 0.001 expressed in scientific notation.

All tests were two-tailed, and the significance level was set at α = 0.05.

## 3. Results

### 3.1. Participants’ Characteristics

The final sample consisted of 101 participants: 48 HCs and 53 MCI patients for demographic and neuropsychological comparisons, according to Albert criteria [26]. Demographic and clinical characteristics are presented in Table 1 and Table 2. The HC group had a mean age of 71.79 years (SD = 5.89) and a mean education of 13.65 years (SD = 3.36). The MCI group had a mean age of 74.58 years (SD = 6.52) and a mean education of 12.09 years (SD = 4.45). Global cognitive screening (MMSE) revealed mean scores of 28.29 (SD = 1.99) for HCs and 27.61 (SD = 1.98) for MCI. See Table 1 and Table 2 for significant differences in socio-demographic and neuropsychological tests.

### 3.2. Group Differences in PIT Performance

To examine the effect of Group on scene interpretation performance (measured by PIT 360°_completion time) while controlling for age, hierarchical linear regression was conducted. PIT 360°_completion time was log-transformed to meet normality assumptions. Model 1 included Group as the sole predictor, while Model 2 added Age as a covariate (see Table 1, MCI participants are older than HCs). The significance of adding Age to the model was evaluated using a likelihood ratio test comparing the nested models.

A hierarchical linear regression was performed to predict log-transformed PIT 360°_completion time scores. In Model 1, Group was entered as the sole predictor. Model 2 added Age as a covariate. The addition of Age significantly improved model fit, Δ*F* (1, 97) = 9.41, *p* = 0.003, accounting for additional variance in PIT 360°_completion time scores beyond that explained by Group alone.

In Model 2, both predictors were significant. Age was significantly associated with PIT 360°_completion time, F (1, 97) = 9.41, *p* = 0.003, with older participants requiring more time to complete the task. Critically, the effect of Group remained significant after controlling for Age, F (1, 97) = 8.59, *p* = 0.004, with MCI patients showing significantly slower performance (M = 92.6 s, SD = 61.9) compared to HCs (M = 65.3 s, SD = 63.3), independent of age differences between groups (Figure 3A). A hierarchical linear regression was performed to predict the squared-root PIT 360°_number of elements scores. In Model 1, Group was entered as the sole predictor. Model 2 added Age as a covariate. Models did not differ significantly (*p* = 0.579). In both models, neither Age nor Group was significant.

Since the only significant result was on PIT 360°_completion time, we proceeded with the subsequent analyses on this variable alone.

We also performed a secondary analysis on a sub-sample of MCI individuals with borderline and pathological scores at the MFPT-UD, which taps visuospatial processes mediated by executive functions.

A Kruskal–Wallis test was conducted due to sample size imbalances to examine differences in PIT completion time scores across three groups: HCs, MCI without executive dysfunction (MCI), and MCIdys. The test revealed significant group differences, χ^2^(2) = 8.24, *p* = 0.016.

Descriptive statistics showed that the HC group had a median PIT 360°_completion time of 40 s (*IQR* = 81.5, *n* = 48), the MCI group had a median of 64 s (*IQR* = 132, *n* = 34), and the MCIdys group had a median of 105 s (*IQR* = 69.2, *n* = 18).

Post hoc pairwise comparisons using Wilcoxon rank sum tests with the Bonferroni correction indicated that the MCIdys group performed significantly worse than HCs (*p* = 0.017, Figure 3B). However, the MCI group did not differ significantly from either HC (*p* = 0.345) or the MCIdys group (*p* = 0.456), suggesting intermediate performance. These findings indicate that executive dysfunction in MCI is associated with significantly slower performance on the PIT 360° task compared to HCs.

### 3.3. Predictive Validity: PIT Performance as a Classifier of MCI Status

Binary logistic regression analyses were conducted to examine whether PIT 360°_completion time scores predicted MCI group membership (coded as 1 = MCI, 0 = HC). A hierarchical approach was used, with Model 1 including PIT 360°_completion time as the sole predictor and Model 2 adding Age as a covariate.

The addition of Age in Model 2 did not significantly improve model fit, Δχ^2^(1) = 2.14, *p* = 0.144, suggesting that Age did not account for additional variance in group membership beyond that explained by PIT 360°_completion time alone. We proceeded to analyze the results of Model 1.

A binary logistic regression was performed to predict MCI group membership from PIT 360°_completion time scores. Model 1, which included only PIT 360°_completion time as a predictor, was statistically significant, χ^2^(1) = 4.70, *p* = 0.030. Higher PIT 360°_ completion time scores were associated with increased odds of MCI classification (β = 0.007, *SE* = 0.003, *p* = 0.034). Specifically, for each one-unit increase in PIT 360°_completion time, the odds of being classified as MCI increased by a factor of 1.01 (see Table 3).

### 3.4. PIT 360° Diagnostic Accuracy and Classification

ROC curve analysis was conducted to evaluate the discriminative ability of the logistic regression Model 1. The model demonstrated an AUC of 0.64 (95% CI [0.53, 0.75]), above the failure cut-off of 0.60 [28]. Using Youden’s index to determine the optimal classification threshold, a predicted probability cut-off of 0.43 was identified, corresponding to a PIT 360°_completion time score of 22 s. The decision rule derived from this analysis suggests that PIT 360°_completion time scores ≥ 22 s indicate an increased likelihood of MCI classification, whereas scores < 22 s suggest a healthy cognitive status (see Figure 4).

At this threshold, the logistic regression model achieved an overall classification accuracy of 65.7% (95% CI [55.4%, 74.9%]), which was significantly better than the no information rate of 52.5%, *p* = 0.006. This indicates that the model’s predictive performance exceeded chance-level classification, based solely on predicting the most prevalent class.

The model demonstrated high sensitivity (86.5%) but modest specificity (42.6%). This asymmetric classification profile reflects a deliberate trade-off with important clinical implications. The high sensitivity (86.5%) but modest specificity (42.6%) indicates that the PIT 360° is particularly effective as a screening tool for ruling out clinically significant executive dysfunction. A negative test result (PIT completion time < 22 s) provides reasonable confidence that an individual does not have executive impairment, as evidenced by the negative predictive value of 74.1%.

However, important caveats apply to this interpretation. First, age significantly influences completion time independent of MCI status, so age-specific norms would improve individual-level interpretation. Second, a negative result indicates intact visuospatial executive functions but does not exclude other cognitive deficits (e.g., memory, language). Third, the 22 s cut-off was derived from this sample and requires validation in independent cohorts before clinical implementation. The positive predictive value was 62.5%, meaning that approximately two-thirds of cases screening positive were true MCI cases, while one-third represented false positives (HC misclassified as MCI). This lower specificity means that positive results require confirmatory evaluation through a comprehensive neuropsychological assessment. The prevalence of MCI in the sample was 52.5%. The model’s detection rate was 45.5%, correctly identifying 45 of the 52 MCI cases. The detection prevalence (proportion classified as MCI) was 72.7%, reflecting the model’s tendency to prioritize sensitivity over specificity. Balanced accuracy, which accounts for class imbalance, was 64.6%. Therefore, while promising for efficient screening in routine practice, the PIT 360° should complement rather than replace comprehensive neuropsychological assessment when executive dysfunction is suspected based on clinical history or functional decline. Cohen’s kappa coefficient was 0.30, indicating fair agreement between predicted and observed classifications beyond chance. McNemar’s test was significant (*p* = 0.001), suggesting asymmetry in misclassification errors between the two groups.

The model demonstrated high sensitivity (86.5%) but low specificity (42.6%), indicating that it correctly identified the majority of MCI cases while frequently misclassifying HC as MCI. The positive predictive value was 62.5%, meaning that 62.5% of cases predicted as MCI were true MCI cases. The negative predictive value was 74.1%, indicating that 74.1% of cases predicted as HCs were indeed healthy.

The prevalence of MCI in the sample was 52.5%. The model’s detection rate was 45.5%, correctly identifying 45 of the 52 MCI cases. The detection prevalence (proportion classified as MCI) was 72.7%, reflecting the model’s tendency to over-predict MCI status. Balanced accuracy, which accounts for class imbalance, was 64.6%.

### 3.5. Comparative Analysis with Standard Cognitive Screening Tests

We also computed classification metrics for two commonly used brief cognitive screening tests in cognitive aging (MMSE and FAB).

A binary logistic regression was conducted to examine whether the MMSE score could predict group membership (MCI vs. HCs). However, the MMSE was not significant (*p* < 0.101). Consequently, we did not proceed to assess the best cut-off performance as MMSE score is not able to classify participants.

A binary logistic regression was conducted to examine whether the FAB score could predict group membership (MCI vs. HCs). The FAB coefficient was negative and highly significant (β = −0.75, SE = 0.17, z = −4.55, *p* < 0.001), indicating that higher FAB scores were associated with decreased odds of MCI classification. The odds ratio was 0.47 (95% CI [0.34, 0.65]), meaning that each one-point increase in FAB score reduced the odds of being classified as MCI by 53%.

ROC curve analysis was conducted to evaluate the discriminative ability of the logistic regression model. The model demonstrated fair discriminative power, with an AUC of 0.79 (95% CI [0.70, 0.88]).

Using Youden’s index to determine the optimal classification threshold, a predicted probability cut-off of 0.698 was identified, corresponding to a FAB score of 15.05. The decision rule derived from this analysis suggests that FAB scores ≤ 15.05 indicate an increased likelihood of MCI classification, whereas scores > 15.05 suggest a healthy cognitive status.

At this threshold, the logistic regression model achieved an overall classification accuracy of 74.0% (95% CI [64.3%, 82.3%]), which was significantly better than the no information rate of 52.0%, *p* < 0.001. This indicates that the model’s predictive performance exceeded chance-level classification, based solely on predicting the most prevalent class.

Cohen’s kappa coefficient was 0.49, indicating moderate agreement between predicted and observed classifications beyond chance. McNemar’s test was significant (*p* < 0.001), suggesting asymmetry in misclassification errors between the two groups. 

### 3.6. Correlations

We performed Pearson’s correlations among neuropsychological tests and PIT 360°_completion time in the entire sample (*N* = 101). PIT 360°–seconds showed significant negative correlations with several neuropsychological measures, indicating that longer PIT 360° performance times were associated with poorer cognitive performance.

Specifically, PIT 360°_completion time correlated significantly with SF (*r* = −0.28, *p* = 0.006), ROCF-Copy (*r* = −0.27, *p* = 0.007), ROCF-Immediate recall (*r* = −0.23, *p* = 0.020), ROCF-Delayed recall (*r* = −0.26, *p* = 0.01), MFPT-UD (*r* = −0.25, *p* = 0.012), AF (*r* = −0.24, *p* = 0.018), PM-D (*r* = −0.22, *p* = 0.027), and FAB (*r* = −0.21, *p* = 0.04). A trend toward significance emerged with PM-I (*r* = −0.19, *p* = 0.057). No significant correlations were found with MMSE, DF, CF, CSS, TMT-A, TMT-B, TMT-BA, or PF (all *p* > 0.10).

## 4. Discussion

This study examined the psychometric properties and clinical utility of the PIT 360°, a VR-based assessment of executive functions in older adults with and without MCI. We developed three specific hypotheses based on the results of existing literature: (1) PIT 360° performance would be lower in MCI patients than in HCs; (2) PIT 360° performance would predict the diagnosis of MCI; and (3) PIT 360° performance would be associated with visuospatial executive tests (e.g., visuospatial fluency). 

According to our findings, the PIT 360° effectively distinguishes between patients with MCI and HCs, with MCI patients requiring significantly more time to complete the PIT 360° compared to HCs. Even after adjusting for age, this difference persisted, suggesting that the observed impairment is not due to general age-related slowing but rather to actual visuospatial executive dysfunction. Furthermore, the test shows a significant association with neuropsychological tests of visuospatial executive functioning, as predicted by our third hypothesis. The PIT 360° captures the complex, integrated executive processes described in Luria’s framework of “active visual perception” [13], which requires strategic visual exploration, hypothesis generation and verification, information integration, and conflict resolution through recursive cycles of exploration and hypothesis testing. In MCI, executive dysfunction disrupts these coordinated processes. Beyond perceptual impairment, these deficiencies include executive breakdowns in spatial reasoning and problem-solving. Qualitative error analysis demonstrates that specific executive failures, such as perseverative and incomplete-correlation errors, are hallmarks of the transition from MCI to dementia, reflecting frontostriatal executive dysfunction rather than primary visuoperceptual deficits [37].

Beyond visuospatial executive processes, the PIT 360° also engages social–cognitive abilities essential to scene interpretation. Inferring the presence of a mouse requires understanding the characters’ emotional states (fear) and intentional behaviors (searching), abilities that draw on theory of mind and pragmatic communication skills. While frontal executive dysfunction impairs the systematic visual search and hypothesis-testing processes described by Luria, social cognition deficits could independently affect performance by making it challenging to infer mental states from facial expressions and body language. The interplay between executive and social–cognitive processes in complex scene interpretation warrants further investigation, particularly given emerging evidence that theory-of-mind abilities decline in MCI and predict functional outcomes. Future versions of the PIT 360° could incorporate eye-tracking to distinguish whether prolonged completion times reflect executive search inefficiency or difficulty with social-emotional processing.

Beyond the conventional emphasis on memory impairment, visuospatial executive dysfunction has become a significant aspect of MCI. Corbo and Casagrande [38] conducted a recent systematic review that summarized the results of 73 studies that looked at higher-level executive functions in older adults with MCI and HCs. Their analysis revealed that the majority of studies confirmed significant executive function alterations in MCI, with notable differences across cognitive domains, including planning, reasoning, problem-solving, and fluid intelligence. When considering MCI subtypes, Corbo and Casagrande noted that multiple-domain MCI generally showed more pronounced executive deficits compared to single-domain presentations. Building on this observation and on the visuospatial nature of several executive assessment tools that distinguish between MCI subtypes, our identification of an MCIdys subgroup characterized by visuospatial dysexecutive deficits (assessed via MFPT-UD) may represent a clinically meaningful approach to characterizing heterogeneity within MCI populations.

Visuospatial executive assessment may be particularly useful for characterizing multiple-domain MCI presentations and identifying patients who would benefit most from focused cognitive rehabilitation interventions.

Evidence from neuroimaging suggests that early network dysfunction may be reflected in visuospatial deficits in MCI. For instance, Berente et al. [39] demonstrated that in patients with multiple-domain amnestic MCI, visuospatial skills showed the largest cognitive domain difference compared to HCs, with an effect size larger than that observed for memory deficits. The temporal pole and superior temporal gyrus, two areas essential for higher-order visuospatial integration, showed selective thinning in these patients’ structures. Resting-state fMRI showed distinctive changes in the visuospatial network: compensatory hyperconnectivity in left frontotemporal connections and decreased interhemispheric connectivity between right and left frontotemporal regions. This pattern suggests that the left hemisphere attempts to compensate through stronger local connections as the right hemisphere, which normally dominates visuospatial processing, is impaired. However, this compensation appears insufficient, as evidenced by persistent visuospatial deficits. Taken together, these results highlight that visuospatial impairments in MCI may include both executive control deficits and disrupted network connectivity, potentially becoming clinically significant and prognostically relevant.

The PIT 360° offers significant advantages in this context because it depends on the integrated involvement of multiple executive processes within a naturalistic scenario, rather than traditional executive function tests that evaluate discrete components. The cognitive demands of this ecological test are especially pertinent to rehabilitation because they are similar to those found in real-life scenarios, such as figuring out social situations, solving problems in home environments, and dealing with complicated environments [24,40]. Further versions of the PIT 360° that integrate eye-tracking technology may offer more detailed information about the spatial exploration patterns of patients with MCI, addressing whether extended completion times are due to processing speed impairments or strategic deficiencies (such as chaotic scanning). Furthermore, by recording reaction times at critical decision points, clinicians would be able to distinguish between executive dysfunction and ideomotor slowing, allowing for a more accurate identification of the cognitive processes behind poor performance.

Our results could inform future cognitive rehabilitation research by providing preliminary evidence for patient stratification. Stratification of MCI patients by MFPT performance revealed that those with visuospatial dysexecutive syndrome showed the most pronounced deficits on the PIT 360°. While this subanalysis reduced statistical power due to smaller sample sizes (*n* = 18 MCIdys vs. *n* = 34 MCI without dysexecutive features), it provides preliminary evidence of heterogeneity within the MCI population, with potential implications for rehabilitation planning. Recent data-driven approaches have identified distinct MCI phenotypes with varying combinations of cognitive impairments [41], each potentially requiring tailored interventions. However, we acknowledge that all MCI patients in our sample showed some degree of executive dysfunction compared to controls, and the clinical utility of further stratification requires validation in larger, independent samples. The current findings suggest a hypothesis for future testing: MCI patients with prominent visuospatial dysexecutive features may require more intensive, multimodal rehabilitation targeting these specific functions, whereas those with relatively preserved executive abilities might benefit from different intervention approaches. Future adequately powered studies should examine whether PIT 360° performance can identify MCI subtypes that respond differentially to specific rehabilitation programs, and whether early identification of dysexecutive syndrome predicts differential progression rates or treatment outcomes.

The PIT 360° demonstrated moderate discriminative ability (AUC = 0.64) with an optimal cut-off of ≥22 s, yielding high sensitivity (86.5%) but modest specificity (42.6%). This asymmetric classification profile reflects a deliberate trade-off that has important implications for clinical application. The model prioritized sensitivity over specificity, making it particularly suitable as a screening tool that minimizes false negatives, that is, it rarely misses individuals with genuine executive dysfunction who would benefit from rehabilitation. A negative test result (PIT_sec < 22 s) provides reasonable confidence that an individual does not have clinically significant executive dysfunction. The negative predictive value of 74.1% indicates that approximately three-quarters of individuals who perform well on the PIT 360° are indeed cognitively healthy. This rule-out capacity is the test’s primary clinical strength, enabling clinicians to efficiently identify individuals who do not require extensive neuropsychological evaluation or immediate executive function rehabilitation, thereby reducing unnecessary testing burden and healthcare costs. However, the relatively low specificity (42.6%) and moderate positive predictive value (62.5%) indicate that positive results (PIT_sec ≥ 22 s) require confirmatory evaluation. Approximately 40% of HCs in our sample were misclassified as potentially impaired. The significant McNemar’s test (*p* = 0.001) confirms this asymmetry in misclassification patterns, with HCs being misclassified more frequently than MCI cases. Therefore, individuals screening positive should undergo a comprehensive neuropsychological assessment to confirm the nature and extent of cognitive deficits before initiating resource-intensive rehabilitation interventions.

These psychometric properties suggest a two-stage clinical pathway that maximizes efficiency while maintaining diagnostic accuracy. In the first stage, all at-risk individuals complete the brief PIT 360° assessment. Those scoring below the 22 s threshold can be reassured and scheduled for routine monitoring rather than immediate extensive testing, with high confidence that clinically significant executive dysfunction is absent. Those scoring at or above the threshold proceed to a comprehensive neuropsychological evaluation to characterize their cognitive profile and determine rehabilitation needs.

The PIT 360° offers a unique and complementary profile to current short cognitive screening tools, as indicated by our comparative analysis. Both the PIT 360° and the FAB showed significant discriminative ability, albeit with notably different classification characteristics, whereas the MMSE was unable to predict MCI group membership in our sample. The FAB showed superior overall discriminative power (AUC = 0.79 vs. 0.64) with high specificity (95.8%) but low sensitivity (53.8%), making it excellent for ruling in MCI when positive (PPV = 93.3%) but prone to missing many true cases. In contrast, the PIT 360° prioritized sensitivity (86.5%) over specificity (42.6%), making it more effective for ruling out MCI when negative (NPV = 74.1%) despite a higher rate of false positives. Importantly, the evaluation of various cognitive processes is reflected in these complementary psychometric profiles. Whereas the PIT 360° captures ecologically valid executive processes in a realistic setting, such as active visual perception, scene interpretation, and hypothesis testing in a complex naturalistic context, the FAB assesses executive functions through abstracted, decontextualized tasks like conceptualization, motor programming, and inhibitory control. This fundamental difference in what the tests measure, beyond their discriminative ability, suggests that using both together provides a more comprehensive assessment of executive functioning than either test alone.

The pattern of relationships between neuropsychological measures and PIT 360°_ completion time provides insight into the cognitive architecture underlying task performance and identifies specific areas that rehabilitation interventions should target. Significant negative correlations were observed between PIT 360°_completion time and measures of immediate and delayed visual memory, executive functions, and visuospatial constructional abilities. These connections imply that integrating various cognitive domains is necessary for effective PIT 360° performance, with executive control acting as a coordinating mechanism [42]. The strong association with the MFPT is noteworthy. The MFPT assesses visuospatial fluency, the ability to generate novel visual patterns, a process mediated by executive functions and anatomically linked to right frontal lobe function, as demonstrated in lesion and neurophysiological studies showing selective impairment following right frontal damage [16,43]. Our exploratory analysis revealed that MCI patients with impaired MFPT performance exhibited significantly slower PIT 360°_completion times, suggesting a shared neurocognitive mechanism. Specifically, both tasks require: systematic visual exploration strategies rather than random scanning, working memory for tracking already-examined elements, cognitive flexibility to shift attention between scene components, and self-monitoring to avoid perseverative responses [16]. The central executive is described in modern working memory models as an attentional controller that coordinates information processing across verbal and visuospatial subsystems [42], supporting cognitive functions like updating, inhibition, coordination, and switching [5,42]. According to neuroanatomical and functional research, the central executive comprises a class of top-down functions that regulate simpler processes [42], demonstrating the connection between working memory and cognitive control.

From a rehabilitation perspective, this integration suggests that interventions targeting visuospatial executive control can leverage these interconnections [40]. Strategy training for systematic visual search and working memory exercises with spatial components may improve multiple domains simultaneously by strengthening the coordinating capacity that manages perceptual, mnemonic, and constructional processes during complex tasks [42]. An integrative rehabilitation approach is further supported by the correlation with long-term visual memory (ROCF delayed recall), which suggests that executive deficits may hinder memory consolidation because delayed recall performance reflects executive functioning related to planning rather than pure visual memory [44]. Combined rehabilitation programs addressing both domains may therefore be more effective than isolated training [44], particularly for patients with visuospatial dysexecutive deficits.

Moreover, one potential strategy for treating executive dysfunction in MCI is technology-based cognitive rehabilitation. Technology-based interventions have been shown to significantly improve various cognitive domains, particularly memory and executive function, often with moderate to large effect sizes, according to a recent systematic review [45]. Beyond efficacy, technology-based delivery offers several useful advantages, including improved cost-effectiveness, real-time performance monitoring, and adaptive difficulty adjustment [45]. 

These characteristics align with rehabilitation principles that emphasize individualized treatment planning. Participants are given tasks tailored to their current level of competence, which promotes optimal brain development while preventing disengagement or frustration. Technology-based cognitive interventions face a persistent challenge: cognitive gains often fail to translate into improved daily functioning. According to a systematic review by Ge and colleagues [45] only two studies showed significant effects on daily living activities, despite improvements in cognition. This disparity emphasizes the value of ecological evaluation instruments such as the PIT 360°, which more accurately predict functional outcomes than neuropsychological tests, because it reflects real-world cognitive demands.

While the PIT 360° demonstrates promising screening utility, several practical considerations merit attention for clinical implementation. First, VR hardware acquisition represents an initial investment, with consumer-grade HNDs costing approximately €300–500, though prices continue to decline as technology matures. However, this cost compares favorably with traditional neuropsychological testing, given its brief administration time (under 3 min) and minimal clinician training requirements. Second, clinician training is straightforward; research assistants in this study achieved competency after observing 2–3 administrations, making the tool accessible to diverse healthcare settings. Third, accessibility remains a consideration: our exclusion criteria (uncorrected visual impairment, recurrent vertigo, physical impairments interfering with headset use) eliminated a few candidates, but these restrictions may limit applicability in frail elderly populations or those with significant physical disabilities. Future iterations could explore alternative presentation formats (e.g., large-screen projection with motion tracking) to extend accessibility. Finally, cybersickness occurred in only 1% of our sample, suggesting good tolerability among community-dwelling older adults; however, careful monitoring remains essential, particularly during initial exposures.

### Limitations

Several methodological limitations should be acknowledged. First, the low specificity (42.6%) suggests a comparatively high rate of false positives, despite the PIT 360°’s high sensitivity (86.5%) in identifying executive dysfunction. This implies that rather than being used as a stand-alone diagnostic tool, it is best suited as a first-level screening tool to rule out clinically significant impairment. To confirm executive deficits, patients who test positive should have a thorough neuropsychological evaluation. Second, before being used in clinical settings, the ≥22 s cut-off value determined by ROC analysis needs to be validated in separate samples. To prove its stability and generalizability, cross-validation research in a range of clinical settings and demographics is required. Third, we did not include eye-tracking technology in our evaluation, even though eye movements are theoretically crucial to Luria’s model of active visual perception. It would have been possible to differentiate between patients with and without executive dysfunction by using eye-tracking data to gain important insights into visual exploration patterns, strategic scanning behaviors, and information-gathering strategies. Future PIT 360° versions that include eye-tracking metrics could improve the tool’s diagnostic precision as well as our understanding of underlying cognitive processes. Fourth, we did not incorporate a formal evaluation of theory of mind or pragmatic language skills, both of which are important for understanding complex scenes. In addition to visuospatial executive functions, the PIT 360° requires the capacity to infer characters’ mental states and comprehend social context. Subtle pragmatic language deficits may have affected performance, even though participants with overt language impairments were unlikely to meet the inclusion criteria and would have been detected during verbal task administration. To separate executive from social–cognitive contributions to PIT 360° performance, future research should include validated measures of pragmatic language and social cognition. This would be especially important for creating focused rehabilitation plans and identifying unique cognitive profiles within the MCI population.

Lastly, one participant experienced cybersickness during VR exposure and was unable to complete the PIT 360°, highlighting a known limitation of VR technology in older adults. While the exclusion rate was minimal (1%), future studies should consider strategies to minimize VR-induced discomfort, such as shorter exposure times, gradual acclimatization protocols, or pre-screening for VR tolerance, particularly in clinical populations who may be more susceptible to motion sickness or vestibular issues.

## 5. Conclusions

The present findings demonstrate that the PIT 360° can effectively distinguish between MCI patients and controls, with particular sensitivity in identifying executive dysfunction within the MCI population.

The PIT 360° assesses executive deficits that may go unnoticed by conventional paper-and-pencil tests but have a substantial impact on day-to-day functioning, requiring active visual perception, hypothesis generation, and detail integration in an ecologically valid virtual environment.

The high sensitivity (86.5%) makes this tool particularly valuable for screening purposes in routine clinical practice, enabling rapid identification of patients who require more comprehensive neuropsychological evaluation. Its significant correlations with established measures of executive functions, visual memory, and verbal fluency highlight the multifaceted nature of active visual perception. Importantly, the brief administration time and reduced costs compared to comprehensive neuropsychological batteries enhance its feasibility for widespread clinical implementation.

In addition to its psychometric qualities, the PIT 360° has several benefits over conventional paper-and-pencil neuropsychological tests, making it suitable for everyday clinical use and rehabilitation planning. First, although participants respond verbally, the task elicits only concise, concrete responses (e.g., recognizing the hidden “mouse” in the scene) without requiring the elaborate verbal explanations, proverb interpretations, or timed word generation that characterize traditional executive assessments. This tool’s relatively low linguistic demand makes it more accessible to people with low educational attainment or limited literacy. Second, even in clinical settings, the short administration time (less than three minutes) and minimal examiner instructions enable effective screening, facilitating the quick identification of patients who would benefit from focused visuospatial executive rehabilitation. Third, the immersive VR interface requires minimal verbal instruction and may reduce performance anxiety through its game-like presentation, potentially increasing accessibility for patients with moderate cognitive decline who might struggle with the abstract requirements of traditional tests. However, a formal assessment of user experience and anxiety levels was not conducted in this study and represents an important direction for future research.

The PIT 360° provides clinicians with a crucial diagnostic advantage from the standpoint of rehabilitation: it allows for more focused and efficient intervention planning by accurately identifying visuospatial executive dysfunction without the confusing memory demands present in conventional verbal learning tasks. Patients identified as having intact memory but impaired executive control may benefit most from strategy training and environmental structuring, whereas those with combined deficits may require multicomponent interventions.

The PIT 360° is particularly well-suited for several clinical contexts. First, in primary care settings, it provides efficient executive function screening before specialist referral, helping general practitioners identify patients who require a comprehensive neuropsychological evaluation. Second, in geriatric clinics conducting routine cognitive monitoring of at-risk populations (e.g., subjective cognitive decline, family history of dementia), the brief administration enables regular surveillance without overburdening clinical resources. Third, in rehabilitation centers, it identifies patients who would benefit most from visuospatial executive-focused interventions, enabling targeted treatment planning. Fourth, in research contexts, it provides an ecologically valid outcome measure for cognitive training and rehabilitation studies. Importantly, the PIT 360° is complementary to, rather than a replacement for, comprehensive neuropsychological assessment. Its primary role is as an efficient first-stage screen to identify individuals requiring more detailed evaluation or to monitor cognitive changes over time in already-diagnosed patients.

These features make the PIT 360° a particularly time-efficient, inclusive, and rehabilitation-focused screening tool that may be able to expand executive function evaluation to more clinical populations that are typically underserved by conventional neuropsychological testing.

## Figures and Tables

**Figure 1 healthcare-14-00095-f001:**
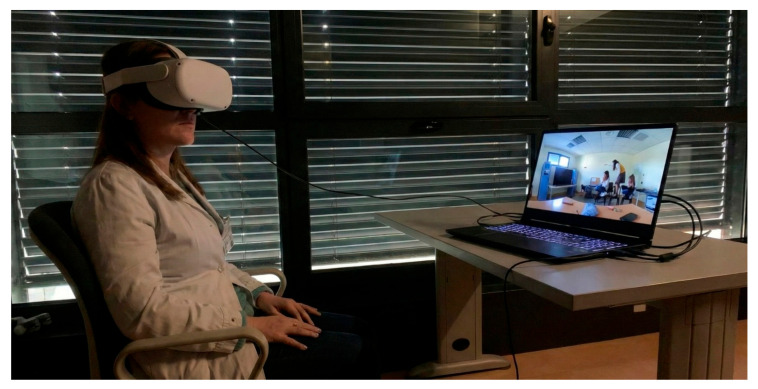
PIT 360° Assessment Setup. A research team member demonstrates the participant configuration during the VR assessment. Participants were seated while wearing an immersive HMD (Oculus Quest 2) and explored the 360-degree environment through head rotation. The monitor shows the real-time VR view for research assistant monitoring.

**Figure 2 healthcare-14-00095-f002:**
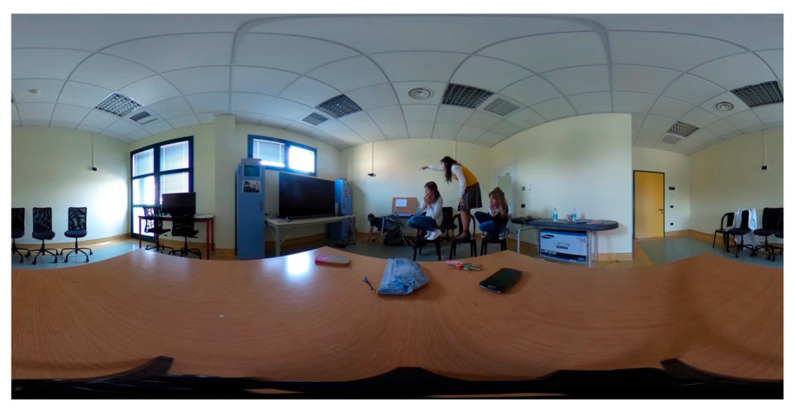
PIT 360° Virtual Environment. Representative 360-degree panoramic view of the virtual domestic scene used in the assessment. Three individuals display fear responses while standing on chairs, and a fourth person searches the floor near furniture. The presence of a hidden threat (mouse) must be inferred from contextual cues, including facial expressions, body postures, and searching behavior. Participants explore this environment through natural head movements while wearing an immersive VR headset. The distorted perspective reflects the 360-degree equirectangular projection format.

**Figure 3 healthcare-14-00095-f003:**
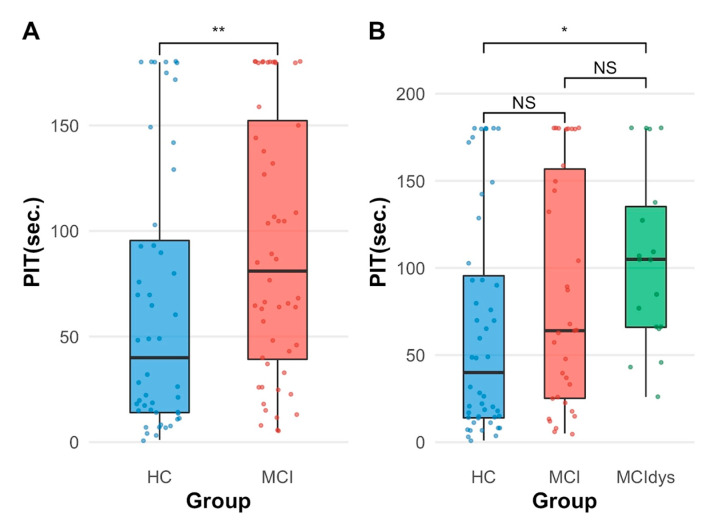
Group differences in PIT completion time. (**A**) Comparison between HCs (*n* = 48) and patients with MCI (*n* = 52). MCI patients showed significantly longer completion times than HCs (*p* = 0.006). (**B**) Comparison across three groups: HCs (*n* = 48), MCI without executive dysfunction (MCI, *n* = 34), and MCI with executive dysfunction (MCIdys, *n* = 18). The MCIdys group performed significantly worse than HCs (*p* = 0.017), while the MCI group showed intermediate performance not significantly different from either HCs (*p* = 0.345) or MCIdys (*p* = 0.456). Box plots represent median (center line), interquartile range (box), and range (whiskers); individual data points are overlaid. * *p* < 0.05, ** *p* < 0.01, NS = not significant.

**Figure 4 healthcare-14-00095-f004:**
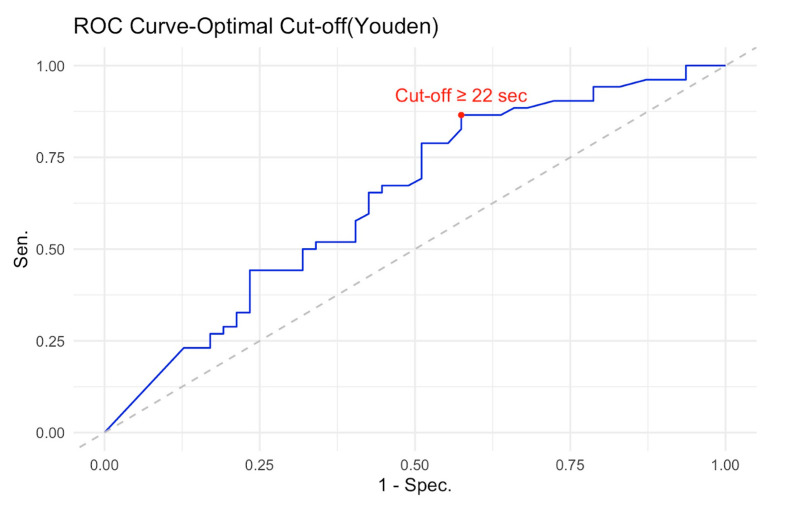
ROC curve for discriminating between MCI patients and HCs using PIT 360°_completion time. The *N* = 100 (48 HCs, 52 MCI). ROC curve demonstrates moderate discriminative ability with an AUC of 0.64 (95% CI [0.53, 0.75]). The optimal cut-off point, determined using Youden’s index (marked with a red circle), was ≥22 s, yielding a sensitivity of 86.5% and a specificity of 43.8%. The diagonal dashed line represents chance-level performance (AUC = 0.50). At the optimal threshold, the model achieved an overall classification accuracy of 65.7% (95% CI [55.4%, 74.9%]), significantly better than chance, *p* = 0.006.

**Table 1 healthcare-14-00095-t001:** Sample demographic and clinical characteristics.

Variables	Overall, *N* = 101	HC, *N* = 48	MCI, *N* = 53	*p*-Value
Sex	NA	NA	NA	>0.9
F	68 (67%)	32 (67%)	36 (68%)	
M	33 (33%)	16 (33%)	17 (32%)	
Age	73.26 (6.35)	71.79 (5.89)	74.58 (6.52)	0.026
Education	12.83 (4.03)	13.65 (3.36)	12.09 (4.45)	0.050
MMSE	27.93 (2.01)	28.29 (1.99)	27.61 (1.98)	0.089
GDS	2.45 (2.54)	2.25 (2.18)	2.62 (2.83)	0.5
ADL	5.91 (0.38)	5.94 (0.24)	5.89 (0.47)	0.5
IADL	7.85 (0.64)	7.96 (0.20)	7.75 (0.85)	0.10
TFI	4.05 (2.53)	3.62 (2.01)	4.43 (2.89)	0.10

Values are presented as *n* (%) for categorical variables and Mean (SD) for continuous variables. HC = Healthy Controls; MCI = Mild Cognitive Impairment; MMSE = Mini-Mental State Examination; GDS = Geriatric Depression Scale; ADL = Activities of Daily Living; IADL = Instrumental Activities of Daily Living; TFI = Tilburg Frailty Indicator. Statistical comparisons conducted using chi-square test for categorical variables and Welch’s *t*-test for continuous variables.

**Table 2 healthcare-14-00095-t002:** Neuropsychological Test Performance.

Cognitive Domain/Test Value	Overall, *N* = 101	HC, *N* = 48	MCI, *N* = 53	*p*-Value
Corsi Forward	4.94 (1.06)	5.21 (1.10)	4.69 (0.97)	0.014
Missing	2	0	2	
Digit Forward	5.98 (1.11)	6.34 (0.91)	5.66 (1.18)	0.002
Corsi Supra-Span	15.31 (7.48)	18.46 (6.80)	12.34 (6.91)	<0.001
Missing	2	0	2	
Trail Making Test-A	33.19 (20.10)	28.60 (15.07)	37.34 (23.12)	0.026
Trail Making Test-B	62.61 (60.16)	43.38 (35.97)	80.04 (71.69)	0.001
Trail Making Test B-A	34.31 (49.55)	18.10 (22.56)	48.98 (61.65)	0.001
Rey-Osterrieth Complex Figure-Copy	31.89 (4.25)	33.43 (1.89)	30.49 (5.23)	<0.001
Rey-Osterrieth Complex Figure-Immediate	17.07 (6.38)	19.09 (5.98)	15.25 (6.24)	0.002
Missing	2	1	1	
Rey-Osterrieth Complex Figure-Delayed	16.77 (6.33)	18.68 (5.72)	15.01 (6.40)	0.003
Missing	1	0	1	
Prose Memory-Immediate	5.12 (1.90)	5.79 (1.30)	4.51 (2.15)	<0.001
Prose Memory-Delayed	5.10 (1.85)	5.84 (1.13)	4.42 (2.11)	<0.001
Frontal Assessment Battery	15.77 (2.16)	16.95 (1.12)	14.70 (2.32)	<0.001
Modified Five Point Test-Unique Designs	23.58 (9.35)	28.41 (7.41)	19.20 (8.80)	<0.001
Phonemic Fluency	36.40 (11.48)	42.79 (8.66)	30.62 (10.66)	<0.001
Semantic Fluency	48.28 (9.45)	52.83 (8.82)	44.15 (8.04)	<0.001
Alternating Phonemic/Semantic Fluency	31.29 (12.90)	38.91 (11.35)	24.39 (10.06)	<0.001

Values are presented as Mean (SD). HC = Healthy Controls; MCI = Mild Cognitive Impairment. Statistical comparisons conducted using Welch’s *t*-test for continuous variables.

**Table 3 healthcare-14-00095-t003:** Logistic Regression Model Predicting MCI Group Membership from PIT Completion Time.

Predictors	OR	95% CI	*p*-Value
PIT 360° (sec.)	1.01	1.00, 1.01	0.034
N° Obs.	100		
AIC	138		
BIC	143	*N*	

OR = Odds Ratio; CI = Confidence Interval; AIC = Akaike Information Criterion; BIC = Bayesian Information Criterion. *N* = 100 participants (HC = 48, MCI = 52).

## Data Availability

The data supporting the findings of this study are available on request from the corresponding author in Zenodo at https://zenodo.org/records/17286463 (accessed on 29 December 2025).

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
