# Peer review of "The Picture Interpretation Test 360°: A Virtual Reality Screening Tool for Executive Dysfunction and Rehabilitation Stratification in Mild Cognitive Impairment"

_healthcare, 2025, doi:10.3390/healthcare14010095_

Round 1
Reviewer 1 Report
Comments and Suggestions for Authors
Dear Authors, I enjoyed reading your manuscript and learning more about the PIT task. Overall, I think the paper is good and the research is solid. There are a few comments that I would like to make to improve the manuscript.
There are two important points that I believe the authors need to address:
1) Why did the authors choose to perform the Picture Interpretation Test in an immersive environment? I couldn't understand the benefits of doing this task in 360 degrees and in a virtual world, as opposed to just showing the picture as originally described. I understand the need to modernize the task for current times, but couldn't the authors just use a new frame and show it in the traditional way? I am an enthusiast for VR and ecological tasks, but I think the authors did not explain the benefits of using this technology for this particular task. I look forward to reading more detailed information and the rationale behind this decision in the manuscript.
2) Why was language not formally assessed? The description of a picture, and understanding the overall scene, as well as inferring the mood of the displayed characters, does require executive functioning and visual attention, but also pragmatic language, theory of mind, and verbal expression. The description of the "Cookie theft" picture has long been used to assess language in Neurology, for example. So, both social/pragmatic language and verbal expression abilities should have been assessed as well. The authors can argue that verbal expression deficits could have been identified informally during the assessment of other cognitive domains, and maybe pragmatic language deficits could have been investigated in the medical history. These are, of course, limitations that should be addressed in the manuscript.
In addition, there are also some minor aspects that I would like to comment:
1) In the introduction, at the end of page 3, the authors were reviewing the history of the PIT task (which was very interesting to read), but suddenly mentioned PIT 360 without explaining how the 360 aspect started. Were there previous papers using 360? Did other researchers use the same frame as the one being used in this paper? This will also help the authors to introduce the fundamentals behind using a 360 approach and the immersive environment.
2) In the results, why combine demographic with NP results? It is usually better to present demographics in table 1, determining no significant group differences in age, sex, educational level, and maybe overall cognitive performance and NP results in another table, maybe even together with the PIT performance. And please try not to use so many initials in the table, makes the reader go back and forth.
3) I didn't find the subanalysis of MCIdys vs MCI adds much to this paper or the discussion. The authors demonstrated that executive functioning deficits influenced the difference between MCI and HC, they already differed as a group in the MFPT scores and the PIT time correlated with other executive function traditional tests. MCI patients usually have some level of executive dysfunction, and further separating this group into subgroups doesn't make much sense from a clinical point of view and decreases statistical power.
4) In the Discussion, much emphasis is placed on the role of visoperception but little is said about the interpretation of social context, which is highly required in this task. In addition, I would be more cautious when affirming that the task is good at ruling out executive dysfunction based on the ROC curve analysis. As the authors demonstrated, age also influences the time for task completion, so on an individual basis, it could be misleading to to rule out executive dysfunction solely on performing the task under the cut-off point. It is an interesting result, but probably not enough to translate into clinical use yet.
Author Response
Dear Reviewer,
We sincerely thank you for your thoughtful and constructive review of our manuscript.
Your comments have substantially improved the clarity and scientific rigor of our work. Below, we provide point-by-point responses to each of your concerns.
Major Concerns:
1) Why did the authors choose to perform the Picture Interpretation Test in an immersive environment? I couldn't understand the benefits of doing this task in 360 degrees and in a virtual world, as opposed to just showing the picture as originally described. I understand the need to modernize the task for current times, but couldn't the authors just use a new frame and show it in the traditional way? I am an enthusiast for VR and ecological tasks, but I think the authors did not explain the benefits of using this technology for this particular task. I look forward to reading more detailed information and the rationale behind this decision in the manuscript.
Response: We agree that this critical point required a better explanation. We have added a new paragraph in the Introduction (page 3, after the description of Rosci's work) that explicitly articulates why the 360° VR format enhances the assessment beyond simply showing an updated picture. The key advantages include: (a) requiring active spatial exploration that mirrors real-world visual behavior and adds executive load absent in static presentations; (b) enabling precise behavioral monitoring of exploration strategies; and (c) increasing ecological validity by simulating real-life scene interpretation demands where relevant information is not confined to a single visual field. We have also added a transition paragraph explaining the evolution from the original PIT to the VR adaptation.
2) Why was language not formally assessed? The description of a picture, and understanding the overall scene, as well as inferring the mood of the displayed characters, does require executive functioning and visual attention, but also pragmatic language, theory of mind, and verbal expression. The description of the "Cookie theft" picture has long been used to assess language in Neurology, for example. So, both social/pragmatic language and verbal expression abilities should have been assessed as well. The authors can argue that verbal expression deficits could have been identified informally during the assessment of other cognitive domains, and maybe pragmatic language deficits could have been investigated in the medical history. These are, of course, limitations that should be addressed in the manuscript.
Response: This is a significant limitation that we failed to adequately address. We have now: (a) added explanation in the Methods section describing how language abilities were informally monitored during verbal task administration; (b) substantially expanded the Limitations section to acknowledge that pragmatic language and theory of mind were not formally assessed, despite contributing to scene interpretation; and (c) added a new paragraph in the Discussion explicitly addressing social-cognitive processes (theory of mind, emotional inference) that interact with executive functions during PIT 360° performance. We agree that this represents a limitation and have proposed specific assessment tools for future studies.
Minor Points:
1) In the introduction, at the end of page 3, the authors were reviewing the history of the PIT task (which was very interesting to read), but suddenly mentioned PIT 360 without explaining how the 360 aspect started. Were there previous papers using 360? Did other researchers use the same frame as the one being used in this paper? This will also help the authors to introduce the fundamentals behind using a 360 approach and the immersive environment.
Response: we have added a transition paragraph explaining the development of the 360° VR version and clarifying that this adaptation was introduced by previous research groups (Realdon et al., 2019; Serino et al., 2017) to enhance ecological validity while preserving Luria's theoretical framework.
2) In the results, why combine demographic with NP results? It is usually better to present demographics in table 1, determining no significant group differences in age, sex, educational level, and maybe overall cognitive performance and NP results in another table, maybe even together with the PIT performance. And please try not to use so many initials in the table, makes the reader go back and forth.
Response: We have reorganized the results presentation by separating demographic data (Table 1) from neuropsychological performance (Table 2). We have also reduced acronym density by including full test names in table columns.
3) I didn't find the subanalysis of MCIdys vs MCI adds much to this paper or the discussion. The authors demonstrated that executive functioning deficits influenced the difference between MCI and HC, they already differed as a group in the MFPT scores and the PIT time correlated with other executive function traditional tests. MCI patients usually have some level of executive dysfunction, and further separating this group into subgroups doesn't make much sense from a clinical point of view and decreases statistical power.
Response: We appreciate your concern about statistical power and clinical utility. We have reduced the emphasis on this subanalysis and reframed it as preliminary evidence of MCI heterogeneity requiring validation in larger samples. We acknowledge that all MCI patients showed executive dysfunction relative to controls, and we have clarified that the clinical value of further stratification remains to be confirmed.
4) In the Discussion, much emphasis is placed on the role of visoperception but little is said about the interpretation of social context, which is highly required in this task. In addition, I would be more cautious when affirming that the task is good at ruling out executive dysfunction based on the ROC curve analysis. As the authors demonstrated, age also influences the time for task completion, so on an individual basis, it could be misleading to to rule out executive dysfunction solely on performing the task under the cut-off point. It is an interesting result, but probably not enough to translate into clinical use yet.
Response: we have added a new Discussion paragraph addressing social-cognitive processes in scene interpretation, acknowledging the interplay between executive and theory of mind abilities. We have also moderated our claims about ruling-out executive dysfunction, adding important caveats regarding age-specific norms, validation needs, and the complementary (not replacement) role of the PIT 360° alongside comprehensive assessment.
We believe these revisions have substantially strengthened the manuscript and addressed all your concerns. Thank you again for your valuable feedback.
Sincerely,
Chiara Stramba-Badiale
Reviewer 2 Report
Comments and Suggestions for Authors
This is a well-designed and relevant study that makes a valuable contribution to ecological and technology-based cognitive assessment in MCI. Please consider minor edits:
Abstract and Title: For the title, recommend removal of "PIT 360° as a Tool" to enhance "first look" clarity of the materials used. Perhaps "virtual reality-based tools" is a more appropriate replacement.
Line 26: ROC, please write out the acronym or remove, or write as the procedure to determine discriminative ability instead of the acronym.
Line 33: (≥22 seconds) is presented as definitive; please clarify that it is preliminary and sample-specific.
Intro: Strong rationale for focusing on visuospatial executive functions in MCI. But, intro could benefit from some condensing, particularly in the historical and theoretical sections, to then free up some room in this area that needs more clarification:
Line 92: Virtual reality introduction needs a bit more detail what types of VR: non-immersive desktop based, non-immersive projected environments, or immersive headsets? It wasn’t until the Methods section Line 242 that I realized the application was immersive and used a headset. Some operational definitions and referencing are required here when introducing your terminology of VR. Then, in Line 102, more description of the device beyond a “realistic three-dimensional scene that they must actively explore…” should be explained, as I was reading the PIT 360 as a 360° immersive view on a smartphone through a gyroscope-controlled view. So describing the set up with a headset viewer is necessary here.
Lines 127 -155 could be condensed, and if any clinical usability (utility highlighted but was this usable and easy to use for the testers?) and safety was examined in the preliminary studies of PIT 360.
Methods
Line 182: Please outline recruitment procedures which is related to sampling.
Line 197 and Line 204: Please list this cut off score of the MMSE of “normal” for transparency.
Line 205: The post-hoc stratification of MCI patients based on MFPT performance should be labeled as exploratory.
Line 243: Please specify if this scene is a “still shot” 360 picture, or a video playing on a loop of a boy searching for something on the floor. Some visual figure here (the VR system itself and the set up of the participant, were they sitting or standing? Did the VR headset require hand controllers to zoom in or out?) and/or the visual scene environment displayed would enhance clarity and set-up repeatability of the methods.
Results: Results are clearly structured with control for age being appropriate. Diagnostic accuracy metrics are comprehensively reported and interpreted.
Discussion: PIT 360 as a screen vs. diagnostic utility is addressed.
Discussion is a bit long with some repetition from earlier of the theoretical frameworks, which likely just need a brief re-referencing.
Again related to the set up description as needing more detail earlier, more attention here should be given to potential implementation barriers, including VR acquisition and cost of devices in settings, clinician training, and accessibility, including the exclusion of individuals with physical impairments that might interfere with VR use may limit real-world generalizability, especially in older adult populations, which is the target.
Conclusion: Concluding with the statement of PIT 360 as a rapid screening tool is appropriate, however some language is strong as the device is likely complementary rather than should replace other tests. Some claims in Line 279, “of the immersive VR interface's easy-to-use, game-like design, which also lowers performance anxiety” are overarching without qualitative open-ended quotes or usability ratings in the data collection and results.
For clinical implications, consider briefly summarizing who would benefit most from PIT 360° screening.
Author Response
Dear Reviewer,
Thank you for your thorough and constructive review. Your suggestions have substantially improved the methodological clarity and clinical applicability of our manuscript.
Below we address each point systematically.
Comments: Abstract and Title: For the title, recommend removal of "PIT 360° as a Tool" to enhance "first look" clarity of the materials used. Perhaps "virtual reality-based tools" is a more appropriate replacement.
Response: Title modification. We appreciate the suggestion to enhance clarity regarding the technology employed. We have revised the title to: “The Picture Interpretation Test 360°: A Virtual Reality Screening Tool for Executive Dysfunction and Rehabilitation Stratification in Mild Cognitive Impairment”. This revision: (1) explicitly identifies the virtual reality-based nature of the assessment for immediate reader comprehension, (2) maintains focus on screening utility while (3) emphasizing the tool's implications for rehabilitation planning and patient stratification, which represents a key contribution of our work as demonstrated by the MCIdys subgroup analysis. We believe this better captures both the technological innovation and clinical utility of our approach.
Comments: Line 26: ROC, please write out the acronym or remove, or write as the procedure to determine discriminative ability instead of the acronym.
Response: We have spelled out "Receiver operating characteristic (ROC)" at first mention in the abstract.
Comments: Intro: Strong rationale for focusing on visuospatial executive functions in MCI. But, intro could benefit from some condensing, particularly in the historical and theoretical sections, to then free up some room in this area that needs more clarification: Line 92: Virtual reality introduction needs a bit more detail what types of VR: non-immersive desktop based, non-immersive projected environments, or immersive headsets? It wasn’t until the Methods section Line 242 that I realized the application was immersive and used a headset. Some operational definitions and referencing are required here when introducing your terminology of VR.
Response: We have added "preliminary, sample-derived" before "cut-off" and included a statement that validation in independent samples is required. Introduction: 4. VR operational definitions (Line 92): We have added a paragraph explicitly defining VR types (non-immersive vs. immersive HMDs) and clarifying that our study employs immersive head-mounted displays. This provides the necessary context before describing the PIT 360°.
Comments: Then, in Line 102, more description of the device beyond a “realistic three-dimensional scene that they must actively explore…” should be explained, as I was reading the PIT 360 as a 360° immersive view on a smartphone through a gyroscope-controlled view. So describing the set up with a headset viewer is necessary here.
Response: we have substantially expanded the description to clarify: (a) the device is an immersive HMD, (b) participants are seated, (c) exploration occurs through head movements only (no hand controllers), and (d) the stimulus is a 360-degree panoramic still image (not video). This addresses confusion about the presentation format.
Comments: Lines 127 -155 could be condensed, and if any clinical usability (utility highlighted but was this usable and easy to use for the testers?) and safety was examined in the preliminary studies of PIT 360.
Response: We have condensed the Realdon and Serino study descriptions from two lengthy paragraphs into one concise paragraph, highlighting key findings and noting safety/usability data from these preliminary studies.
Comments: Methods: 7. Line 182: Please outline recruitment procedures which is related to sampling.
Response: We have added a detailed description of recruitment pathways and diagnostic procedures.
Comments: Line 197 and Line 204: Please list this cut off score of the MMSE of “normal” for transparency.
Response: We have specified the Italian age- and education-adjusted MMSE cut-offs used for inclusion/exclusion.
Comments: Line 205: The post-hoc stratification of MCI patients based on MFPT performance should be labeled as exploratory.
Response: We have explicitly labeled the MCIdys stratification as "an exploratory post-hoc analysis”.
Comments: Line 243: Please specify if this scene is a “still shot” 360 picture, or a video playing on a loop of a boy searching for something on the floor. Some visual figure here (the VR system itself and the set up of the participant, were they sitting or standing? Did the VR headset require hand controllers to zoom in or out?) and/or the visual scene environment displayed would enhance clarity and set-up repeatability of the methods.
Response: We have substantially expanded Section 2.3 to include: (a) specific HMD model, (b) detailed setup procedure (seated, adjustment, no controllers), (c) clarification that the stimulus is a still 360° image (not video), (d) monitoring procedures. We have also added Figure 1 and Figure 2 showing virtual environment and virtual setup.
Comments: Discussion is a bit long with some repetition from earlier of the theoretical frameworks, which likely just need a brief re-referencing.
Response: We have reduced redundancy by condensing the Luria framework re-explanation from ~150 to ~50 words, with brief reference back to the Introduction rather than full re-explanation.
Comments: Again related to the set up description as needing more detail earlier, more attention here should be given to potential implementation barriers, including VR acquisition and cost of devices in settings, clinician training, and accessibility, including the exclusion of individuals with physical impairments that might interfere with VR use may limit real-world generalizability, especially in older adult populations, which is the target.
Response: We have added a new "Implementation Considerations" section addressing: VR acquisition costs (~€300-500), minimal training requirements, accessibility limitations (physical impairments), and cybersickness rates (1%). This provides realistic guidance for clinical adoption.
Comments: Conclusion: Concluding with the statement of PIT 360 as a rapid screening tool is appropriate, however some language is strong as the device is likely complementary rather than should replace other tests. Some claims in Line 279, “of the immersive VR interface's easy-to-use, game-like design, which also lowers performance anxiety” are overarching without qualitative open-ended quotes or usability ratings in the data collection and results.
Response: We have revised claims about "easy-to-use " and "lowers performance anxiety" to acknowledge these were not formally assessed and require future investigation.
Comments: For clinical implications, consider briefly summarizing who would benefit most from PIT 360° screening.
Response: We have added a paragraph specifying clinical contexts where PIT 360° is most appropriate (primary care, geriatric clinics, rehabilitation centers, research) and explicitly stating it is complementary to, not a replacement for, comprehensive assessment.
We believe these revisions have substantially enhanced methodological transparency and clinical utility.
Thank you again for your valuable feedback.
Sincerely,
Chiara Stramba-Badiale